# Powassan Virus Induces Structural Changes in Human Neuronal Cells In Vitro and Murine Neurons In Vivo

**DOI:** 10.3390/pathogens11101218

**Published:** 2022-10-21

**Authors:** Jacob Nelson, Lorenzo Ochoa, Paula Villareal, Tiffany Dunn, Ping Wu, Gracie Vargas, Alexander N. Freiberg

**Affiliations:** 1Department of Pathology, University of Texas Medical Branch, Galveston, TX 77555, USA; 2Biomedical Engineering and Imaging Sciences Group, University of Texas Medical Branch, Galveston, TX 77555, USA; 3Department of Neuroscience, Cell Biology and Anatomy, University of Texas Medical Branch, Galveston, TX 77555, USA; 4Institute for Human Infections and Immunity, University of Texas Medical Branch, Galveston, TX 77555, USA; 5Center for Biodefense and Emerging Infectious Diseases, University of Texas Medical Branch, Galveston, TX 77555, USA; 6Sealy Institute for Vaccine Sciences, University of Texas Medical Branch, Galveston, TX 77555, USA

**Keywords:** powassan virus, west nile virus, human neural stem cell (hNSC)-derived neuron astrocyte co-culture, neuroinflammation, neurovirulence, laminal membrane structures, clarity

## Abstract

Powassan virus (POWV) is a tick-borne flavivirus (TBFV) that can cause severe encephalitis in humans with a case–fatality rate as high as 11%. Patients who survive severe encephalitic disease can develop long-term neurological sequelae that can be debilitating and life-long. In this study, we have sought to characterize a primary human fetal brain neural stem cell system (hNSC), which can be differentiated into neuron and astrocyte co-cultures, to serve as a translational in vitro system for infection with POWV and a comparative mosquito-borne flavivirus (MBFV), West Nile virus (WNV). We found that both viruses are able to infect both cell types in the co-culture and that WNV elicits a strong inflammatory response characterized by increased cytokines IL-4, IL-6, IL-8, TNF-α and IL-1β and activation of apoptosis pathways. POWV infection resulted in fewer cytokine responses, as well as less detectable apoptosis, while neurons infected with POWV exhibited structural aberrations forming in the dendrites. These anomalies are consistent with previous findings in which tick-borne encephalitis virus (TBEV) infected murine primary neurons formed laminal membrane structures (LMS). Furthermore, these structural aberrations are also recapitulated in brain tissue from infected mice. Our findings indicate that POWV is capable of infecting human primary neurons and astrocytes without causing apparent widespread apoptosis, while forming punctate structures reminiscent with LMS in primary human neurons and in vivo.

## 1. Introduction

Powassan virus (POWV) is a tick-borne flavivirus (TBFV) that is classified in the family *Flaviviridae*, genus *Flavivirus* [1,2]. Other arthropod-borne viruses in this genus include viruses such as tick-borne encephalitis virus (TBEV), Langat virus (LGTV) and Omsk hemorrhagic fever virus (OHFV), as well as the mosquito-borne flaviviruses (MBFV), such as yellow fever virus (YFV), West Nile virus (WNV) and Zika virus (ZIKV). POWV is endemic to the United States and is closely related to TBEV, which is found throughout Europe and Asia. The vector of POWV is the ixodid tick with Ixodes scapularis being the primary vector throughout the Eastern United States [3]. There are recent reports of POWV-infected ticks being found in areas that had not previously been included in the range for POWV, specifically Appalachian Virginia, and there is potential for expansion south and west due to the range of *Ixodes scapularis* [4,5]. Between 2011 and 2020, 179 POWV cases were reported to the CDC; however, the number of cases is likely under reported due to the need for increased surveillance [6,7].

POWV can cause encephalitis, which can potentially be lethal (case–fatality rate 11%), has been described to result in long-term neurological sequelae in survivors, including motor dysfunction, cognitive deficits and behavioral changes [2,3,7,8,9,10]. Neurotropic flaviviruses including POWV, TBEV, WNV and ZIKV cause a biphasic course of disease characterized by a widespread infection throughout the body with accompanying inflammatory response followed by a neuroinvasive phase [10,11]. During the latter phase involving the central nervous system, inflammatory cytokines and chemokines combined with intravasation of CD8+ T-cells have been implicated in the pathology [12]. In the acute phase of encephalitis, TNF-α and IL-1β are significant markers for apoptosis and IFN-g has been described as a marker for overall inflammation [12,13]. While direct neuronal damage due to infection with MBFVs is well-documented, reports indicate that TBEV does not cause neuronal cell death from viral infection alone [14]. The lack of neuronal death from viral infection has not yet been investigated for POWV.

Neuronal cell line systems have been primarily used to study the response to flavivirus infection. It has been shown that immortalized SK-N-SH cells release neuroinflammatory cytokines such as IL-1β, IL-4, IL-6 and TNF-α in response to WNV infection, which is similar to elevated cytokine levels detected in human cases [15,16]. Induced pluripotent stem cells also have a strong inflammatory response to WNV infection including upregulation of cytokines IL-1β, IL-4, IL-6 and TNF-α [16]. Infection with TBEV in neuronal cell lines, as well as primary astrocytes, has confirmed that an inflammatory response is elicited [14,17]. These inflammatory responses have also been described in human cases for both WNV and POWV, as well as in animal models [8,11,12,18,19,20].

Previous studies have shown that TBFVs form distinctive structural aberrations in the dendrites of infected primary murine neurons, known as laminal membrane structures (LMS) [21]. These structures comprise virions that are encased in multiple membranes inside the dendrites, resulting in localized swelling of the neuronal projections [21]. The formation of LMS is unique to TBFVs as they are caused by the conserved stem-loop 2 structure of the 5′ untranslated region (UTR) binding to host proteins including FMR1 [22]. This phenomenon is notably absent from MBFVs as there is a considerable difference in the 5′ UTR of MBFVs and TBFVs [22]. The formation of LMS has been shown to cause reduced transport of mRNA used to translate neuron signaling proteins ARC and BDNF from the hillock of the neuron to the synapse, resulting in compromised neuron function. While LMS formation and composition has been demonstrated to develop in murine neurons with TBEV, it has yet to be shown if TBFVs can cause LMS formation in human neurons [22]. Furthermore, there have not been any reports yet for LMS formation in vivo and their potential impact on development of neurological disease.

Studies of viral infections in primary human neurons are desired because of their physiological cellular responses and translational application compared to cell lines. However, limitations exist due to the challenges of working with non-replicative cells that are difficult to obtain. The two alternative systems most utilized are human neuronal cell lines and primary mouse or rat neurons [23,24,25,26]. These cell systems are important for initial studies investigating virus-induced molecular pathogenesis and the evaluation of antiviral therapeutics, but they are limited translationally by interspecies differences and physiological anomalies. Here, we sought to evaluate the applicability of a primary human neuron and astrocyte co-culture system that has been differentiated from human fetal brain neural stem cells [27,28]. This model has been utilized previously to study viral infection of neurons, notably for La Crosse virus (LACV) [23,25,29], Nipah virus [30] and ZIKV [23,25,29]. In addition to translational advantages, this model can produce human primary cells that are cultured rapidly and consistently, while eliminating donor variability. We were able to demonstrate that the hNSC-derived neuron/astrocyte co-culture system is susceptible to infection with both POWV and WNV and support viral replication. In contrast to WNV, infection with POWV elicits a mild inflammatory response, but still induces the release of chemokines that are known to attract activated lymphocytes. We have also shown that POWV infection results in the detection of aberrations in human neuronal dendrites without causing apoptosis, while WNV causes activation of apoptosis with no detection of dendritic aberrations. In addition, POWV-induced aberrations were also found in vivo in infected murine neurons in the brain.

## 2. Materials and Methods

### 2.1. Cell Culture and Viruses

BHK cells were cultured using DMEM (Gibco, Grand Island, NE, USA) supplemented with 10% FBS and 1% Sodium Pyruvate (Gibco, Grand Island, NE, USA), glucose (Gibco, Grand Island, NE, USA) and NEAA vitamin solution (Gibco, Grand Island, NE, USA) at 37 °C and 5% CO_2_. Human neural stem cells (hNSCs) were maintained as neurospheres as previously described [27,28]. Briefly, neurospheres were maintained in DMEM/F12 (Corning, Corning, NY, USA) media supplemented with epidermal growth factor (EGF) (20 ng/mL) (R&D Systems, Minneapolis, MN, USA), fibroblast growth factor (FGF) (20 ng/mL) (R&D Systems, Minneapolis, MN, USA), leukocyte inhibitory factor (LIF) (10 ng/mL) (Chemicon, Kuurne, Belgium), heparin (5 μg/mL) (Sigma-Aldrich, Burlington, VT, USA) and insulin (25 μg/mL) (Sigma-Aldrich) and maintained at 37 °C and 5% CO_2_. hNSCs were adhered to glass coverslips in 24-well plates coated with 0.01% poly-D-Lysine (Sigma-Aldrich, Burlington, VT, USA) and laminin. Differentiation into neuron and astrocyte co-cultures was performed for 10 days and comprised of an approximate equal ratio of each cell type using B2 neurobasal supplemented media [27].

Viruses were provided by the World Reference Center for Emerging Viruses and Arboviruses (WRCEVA) at the University of Texas Medical Branch. The strain of POWV used was the progenitor type 1 strain, LB; while the strain of WNV used was WNV/NY/99.

### 2.2. Infection of Cell Cultures

Infections of hNSC-derived neuron/astrocyte co-cultures were carried out using either 0.1 or 1.0 multiplicity of infection (MOI) with the specified virus by removing media and adding 100 µL of diluted virus to each well, followed by a 1-h incubation at 37 °C and 5% CO_2_. Following infection, cells were rinsed three times with PBS and then replenished with fresh differentiation media. Additionally, negative controls of mock-infected cells were carried out in a similar manner using hNSC differentiation media without the addition of virus in place of inoculum or heat-inactivated virus was substituted for infectious virus (1.0 MOI). Heat inactivation of virus was achieved by placing undiluted virus stocks in a 56 °C water bath for 1 h prior to dilution and inoculation.

### 2.3. Plaque Assay

Viral titers were determined using standard plaque assay with BHK cells. Six-well plates with 250,000 BHK cells per well were infected with ten-fold serially diluted supernatant for 1 h before removing the supernatant, washing 3× with PBS and having the media replaced with overlay media (2× MEM (Gibco, Grand Island, NE, USA), 4% FBS, 2% penicillin/streptomycin mixed with equal parts of 1.6% tragacanth). Plates were incubated for three days before being stained with crystal violet dissolved in 10% formalin.

### 2.4. BioPlex Assay

Supernatant from POWV or WNV infected cells was collected at 0, 4-, 24-, 48- and 72-h post-infection and stored at −80 °C before being γ-irradiated (5 MRad) and transferred to BSL2 conditions. Supernatants were then used to measure cytokine and chemokine levels using the BioPlex human cytokine 27-plex panel (BioRad, Hercules, CA, USA) following manufacturers’ instructions. Values are reported as total amount pg/mL.

### 2.5. Apoptosis Assay

Differentiated hNSC co-cultures were grown in 96-well plates at a density and manner previously described in Section 2.1 and Section 2.2. The cells were then infected with 1.0 MOI of WNV or POWV and heat-inactivated WNV or POWV. A quantity of 10 μM polyinosinic:polycytidylic acid (poly I:C) (Sigma Aldrich, Burlington, VT, USA) was used as positive control. After 72 h, caspase 3/7 cleavage was measured as a marker for apoptosis using the ApoTox-Glo (ProMega, Madison, WI, USA) kit following manufacturers protocols.

### 2.6. Immunofluorescence Staining

Primary hNSC co-cultures were grown on coverslips and infected as previously described in Section 2.1. After 72 h post-infection, supernatant was removed and replaced with 10% formalin overnight. The formalin was removed, and coverslips were rinsed with PBS before being stored in PBS at 4 °C. The PBS was then removed at time of staining and cells were permeabilized with 0.1% Triton X-100 for 5 min then rinsed again three times with PBS. Blocking with 10% normal donkey serum in 1% BSA and PBS for 30 min was used to reduce non-specific binding. Primary antibodies were diluted 1:1000 in 1% BSA and PBS then incubated overnight at 4 °C. The coverslips were then washed three times with PBS and corresponding secondary antibodies were added at a dilution of 1:500 and incubated for 1 h at room temperature. A final rinse with PBS was used before coverslips were mounted onto slides and stored at 4 °C until imaging. Primary antibodies used were rabbit polyclonal anti-POWV (obtained from WRCEVA, UTMB) and was used to label POWV (diluted 1:500); rabbit polyclonal anti-WNV (AbCam-ab25886) was used to label WNV (diluted 1:1000); chicken monoclonal anti-Map2 (AbCam-ab5392) was used to label neurons (diluted 1:1000); goat monoclonal anti-GFAP (AbCam-ab53554) was used to label astrocytes (diluted 1:1000). Secondary antibodies used were donkey anti-rabbit AlexaFluor 594 (AbCam-ab150076), donkey anti-chicken FITC (AbCam-ab63507) and donkey anti-goat AlexaFluor 405 (AbCam-175664) for labelling and imaging (diluted 1:500).

### 2.7. Tissue Clearing and Labeling

BALB/c mice were infected with 1000 pfu of POWV intraperitoneally. Animals were monitored daily for clinical signs of disease and monitoring was increased to twice or three times per day depending on the health scoring observed at previous observations. Brains were collected at time of euthanasia which occurred 8 days post inoculation. Tissues were placed into formalin for inactivation according to institutionally approved protocols, which was a minimum of 21 days, with samples being in formalin up to six months at room temperature.

Brains were removed from formalin after being transferred to a BSL2 setting. They were then cleared using Life Canvas Technologies SMARTCLEAR and SMARTLabel systems and reagents (Life Canvas Technologies, Cambridge, MA, USA). Tissues were imbedded into an epoxy hydrogel matrix using SMARTSHIELD Epoxy and ShieldON solution and were rocked on an orbital rocker for 5 days at 4 °C. Samples were then transferred to ShieldOFF solution for 24 h at 37 °C. Samples were then stored in PBS until being cleared using the SMARTCLEAR system. Settings for the SMARTCLEAR system were 90 V potential, 1500 mA current, 0.1 RPM rotations and 50 °C. The amount of time needed to clear samples varied between 4–6 days. Samples were then either stored in PBS or proceeded to be immunolabeled using the SMARTLabel system, the samples appeared translucent at this point. Hemispheres were placed in the SMARTLabel primary solution overnight and then labeled for 24 h with primary antibody and primary buffer. The settings for the primary antibody labeling were 90 V potential, 1500 mA current, active stirring and 37 °C. The tissues were placed in 50 mL PBS overnight with shaking, changing the PBS three times. Next, the tissues were washed using the SMARTClear with the settings being 90 V potential, 1500 mA current, 0.1 RPM rotations and 37 °C overnight. Secondary antibodies were then added the following day and allowed to incubate for 6 h. Samples were then washed with PBS for six hours and then placed in 4% PFA overnight. The following day they were rinsed in PBS for another six hours and stored in PBS until used for microscopy. Two days before imaging, samples were placed in a solution of 50% EasyIndex and 50% PBS overnight at room temperature on an orbital rocker. The solution was then changed to 100% EasyIndex and incubated overnight at room temperature. Samples appeared visually clear at this point and ready to image. Antibodies used for labeling were as follows: rabbit polyclonal anti-POWV (obtained from WRCEVA, UTMB); chicken monoclonal anti-Map2 (AbCam-ab5392); donkey anti-rabbit AlexaFluor 594 (AbCam- ab150076) and donkey anti-chicken FITC (AbCam-ab63507). All antibodies were used at a concentration of 10 µg per sample, diluted in 4 mL of SMARTclear buffer.

### 2.8. Microscopy and Data Analysis

Tissue samples were imaged using a Prairie Ultima IV (Bruker, Middleton, WI, USA) upright multiphoton microscope. For 2-photon fluorescence microscopy, a 4 × 0.16 N.A. objective with 13 mm working distance (UPLSAPO 4X, Olympus, Tokyo, Japan) and a 40 × 0.8 N.A. objective having a working distance of 3 mm (MRD07420, Nikon, Tokyo, Japan) was used for image collection. Illumination for excitation of fluorescence was provided by a femtosecond laser (Mai Tai, SpectraPhysics, Santa Clara, CA, USA) tuned to 800 nm. Fluorescence was collected using a 2-photon standard filter set, with filter 1 parameters of bandwidth 604 ± 45 nm, filter 2 parameters of bandwidth 525 ± 70 nm and a dichroic mirror cutoff at 575 nm. Samples were mounted on a 30-mm cage plate (CP06, ThorLabs, Newton, NJ, USA) between two #1.5 cover glasses and immersed in 70% TDE (2,2-Thiodiethanol) for optical clearing. Z-stacks were acquired in sequential order of row by row, right and down pattern with ~20% overlap. Three-dimensional stitching reconstructions were generated using ImageJ with a pairwise-stitching algorithm [31] with 25% overlap. Surface analysis using background subtraction and fixed signal thresholding (900–65,535 on 16-bit images) generated 3D reconstructed objects for quantification. Volume and colocalization were obtained from 3D reconstructions. Analysis of the reconstruction and additional 3D images were performed using IMARIS version 9.7.

All co-culture slides were imaged with a Zeiss Inverted LSM 800 confocal microscope using two GaAsP PMT detectors and a motorized stage. Lasers used for excitation were at a wavelength of 405, 488 and 561 nm. Objectives used were Plan-Apochromat 63× 1.4 oil, LD LC1 Plan-Apochromat 40× 1.2 water, Plan-Apochromat 20× 0.8 air, EC Plan-Neufluar 10× 0.45 air, APO 5× 0.25 air. Software used for image capture was Zeiss ZEN 2.3 Blue and images were processed using IMARIS software.

Percent infection calculations were performed using 12 total images at 20× magnification captured across four separate mock-infected slides and 12 images at 20× magnification captured across four separate slides infected with either WNV or POWV. Size of puncta calculations were measured manually using the linear measure tool after image thresholding and segmenting on images across four separate slides of either mock- or POWV-infected cells; ten mock images and ten infected images were captured using 20× magnification. Ten puncta from each image were manually measured using imageJ.

### 2.9. Statistical Analysis

Statistical analysis for differences in apoptosis and viability were performed using an unpaired *t*-test with Welch’s correction. Analysis for viral titers was performed using a paired *t*-test. Comparisons of cytokine production via BioPlex assay were analyzed using a multiple measures ANOVA. Punctum-size comparisons were performed using an unpaired *t*-test with Welch’s correction. All statistical analyses were conducted using GraphPad Prism 9.0.0, *p*-values for all figures are * *p* ≤ 0.05, ** *p* ≤ 0.01, *** *p* ≤ 0.001, **** *p* ≤ 0.0001.

## 3. Results

### 3.1. hNSC-Derived Neuron/Astrocyte Co-Cultures Are Susceptible to Infection with POWV and WNV

Human neurons are difficult to obtain and propagate in culture. However, K048 human neuronal stem cells can be differentiated into neuron and astrocyte co-cultures. Previously, described data showed that the neurons in co-culture are similar to those found in the cortex, as they are both excitatory and inhibitory in function [28]. Astrocytes in the co-culture system are typical for non-activated astrocytes and characterized by expressing astrocyte markers, including glial fibrillary acidic protein (GFAP). The differentiated hNSC-derived neuron/astrocyte co-cultures comprised a ratio of 52.3% neurons and 47.7% astrocytes determined by imaging (Appendix A).

First, we evaluated the susceptibility of hNSC-derived neuron/astrocyte co-cultures to infection with POWV and WNV. Co-cultures were either MOCK-treated or infected with 1.0 MOI of WNV or POWV. After 48 h, WNV-infected co-cultures started to develop cytopathic effect (CPE) characterized by cellular swelling and detachment, while POWV-infected cells still resembled the monolayer of MOCK cells. At 72 h post-infection, POWV-infected co-cultures showed minor swelling in the cellular bodies under bright field microscopy, but still mainly resembled MOCK cells (Figure 1a). Co-cultures infected with WNV on the other hand showed swelling of the cellular bodies and reduced projections as compared to MOCK cells, indicating a higher degree of CPE due to infection (Figure 1a). Using confocal microscopy at 20× magnification 72 h post-infection, the neuron/astrocyte co-cultures revealed infection in both cell types by both viruses (Figure 1b). To determine the replication growth kinetics for POWV and WNV in hNSC-derived neuron/astrocyte co-cultures, supernatant was sampled daily, and virus titer determined by plaque assay (Figure 1c). Both viruses showed similar initial growth kinetics within the first 24 h post infection. For WNV, peak viral titers were reached after 48 h post-infection and remained consistent until 96 h post-infection. No obvious differences in viral titers were measured between MOI 0.1 and 1 throughout the time course. Peak viral titers for POWV were reached after 72 h post-infection. Slightly higher titers were determined for MOI 1 until 48 h post-infection. No differences in titer were detected at 72- and 96-h post-infection. While POWV grew to slightly higher titers, no significant statistical difference to WNV could be detected. The higher viral titers of POWV and plateau of WNV at 48 h could be explained by the higher CPE of WNV compared to POWV. WNV infected 80% of neurons but only 13% of astrocytes, while POWV-infected 59% of neurons and 34% of astrocytes, which could contribute to the trends seen in the replication kinetics (Appendix A). In a separate experiment, POWV was observed to infect neuron/astrocyte co-cultures for an extended time-window of up to 14 days and resulted in a plateaued virus titer without causing any CPE or visual differences compared to MOCK-infected cells (Appendix A). Neuron/astrocyte co-cultures infected with 1.0 MOI of either WNV or POWV appeared viable at 72-h post-infection under bright field microscopy. At 96 h post-infection, WNV-infected cells caused widespread CPE with almost all cells detaching from the monolayer. Due to this observation, future experiments were terminated at 72 h post-infection.

Next, we determined cell viability and apoptosis in POWV- and WNV-infected hNSC-derived neuron/astrocyte co-cultures. Differentiated co-cultures were infected with either 1.0 MOI or 0.1 MOI of either WNV or POWV and virus-induced apoptosis measured via caspase 3/7 cleavage. Two different MOIs were used to ensure that a suitable range for detection of caspase cleavage was achieved. There were no significant changes in the viability. However, there was a significant increase in apoptosis for control cells that were MOCK-infected and treated with 10 μM poly I:C. For both viruses and MOIs, a similar level of cell viability was detectable (Figure 2a). However, an MOI of 1 of WNV caused an elevated level of caspase cleavage, comparable to Poly I:C, while POWV had similar levels of apoptosis to MOCK-infected cells (Figure 2b). When compared to the heat-inactivated non-replicating virus controls, there was no detectable difference from the MOCK-infected cells; thus, changes seen in infected cells are due to virus replication and not infection. This would indicate that in this co-culture system, POWV is not inducing direct cell destruction or apoptosis of infected neurons or astrocytes, while WNV infection causes activation of apoptosis.

### 3.2. POWV Does Not Induce an Inflammatory Response in hNSC-Derived Neuron/Astrocyte Co-Cultures, While WNV Causes Widespread Inflammation

Inflammatory cytokines released by neurons and astrocytes have been noted to play a role in pathogenesis during flaviviral encephalitis [11,12,15,16,32,33,34,35]. After demonstrating that hNS-derived neuron/astrocyte co-cultures are susceptible to infection with either POWV or WNV, we wanted to determine their cellular immune responses to infection. Supernatants from co-cultures infected with 1.0 MOI of either POWV or WNV were collected at 4-, 24-, 28- and 72-h post-infection and tested for translational changes in pro-inflammatory cytokines and chemokines (Figure 3) using the BioPlex human 27-plex cytokine panel. Previously, ability of the co-cultures to respond to an external stimulus was demonstrated [23] and confirmed here using Poly I:C as a positive control. Overall, cellular innate immune responses to POWV infection were weak, with multiple readings below the limit of detection, except for IFN-g and IL-7. Elevated pro-inflammatory cytokine levels were detected for WNV infection at 48- and 72-h post-infection, with key inflammatory cytokines IL-1β, TNF-α, IL-6 and IL-8 showing a statistically significant increase. Chemokine responses were also strong in WNV-infected co-cultures at 48- and 72-h post-infection. Responses to POWV were elevated in several chemokines, most notably VEGF, which was consistent with the levels of chemokines triggered by WNV. This is consistent with the levels of pro-inflammatory cytokines (IL-1β, IL-8, IL-8, TNF-α) produced during WNV infection of the neuronal cell line SK-N-SH [15]. These findings correlate with the apoptosis data for WNV, signifying that an inflammatory response might be driving caspase cleavage, while a reduced inflammatory response might be tied to lower apoptosis, as observed for POWV.

### 3.3. West Nile Virus Causes Destruction of Astrocytes and Neurons

WNV is a neurotropic flavivirus that is responsible for direct neuronal death and inflammation resulting in apoptosis [16]. Although neurons are the primary target, astrocytes can also be infected and are a source of cytokines and chemokines in response to WNV infection [17]. After 72 h post-infection, both neurons and astrocytes showed WNV establishing a productive infection (Figure 4a, Mock images Figure 1b). WNV was found to be isolated to the body of the neurons and astrocytes, while being absent from the dendrites of infected neurons. Large areas of cellular destruction of both neurons and astrocytes were present, as evidenced by ruptured cells and fractured neuronal projections, which was exemplified after images were processed to segment neuron, astrocytes and E-protein signal and reconstructed. Cells were surface-rendered into translucent cells (neurons: green surfaces; astrocytes: purple surfaces) with opaque surface of infected regions stained for E protein (Figure 4b). Under further magnification, WNV is found to be concentrated within inclusions inside of neurons (Figure 4c) and causing fragmentation of neuronal processes. After image processing, it can be seen that the neurons are not fully containing the virus and are in fact lysing (Figure 4d). Increased quantities of cellular debris can be found in these areas from the bodies of infected cells in addition to the damaged neuronal and astrocytic processes. The images shown confirm that WNV is destructive to human neurons and astrocytes in culture.

### 3.4. Powassan Virus Causes Aberrations in Human Neurons In Vitro

While POWV can cause serious CNS disease and has been shown to induce strong inflammatory responses [36], there is currently no evidence that POWV causes direct damage to neurons. Imaging of neuron/astrocyte co-cultures infected with POWV reveal that neurons still show intact axons and dendrites that appear similar to mock controls (Figure 1b and Figure 5a). At lower magnifications (20×), no cellular debris or apoptotic bodies were observed, and neuronal processes are formed and unfragmented, despite the presence of viral antigens in both neurons and astrocytes. At higher magnification (63×), two apparent differences are detected in the distribution of POWV as compared to WNV: (i) aberrations are forming in the dendrites of infected neurons and (ii) POWV is present in dendrites, while WNV was confined to the neuronal bodies (Figure 4a and Figure 5b). The spread into dendrites perpetuates the formation of aberrations in the dendrites as the swelling of the dendrites is caused by areas of virus located within these anomalies (Figure 5c). It appears that POWV is not causing obvious CPE and after image processing, it can be visualized that POWV is contained within the cells, especially dendrites (Figure 5d). Further isolation of the swollen dendrite areas highlights that the presence of POWV results in the formation of distinct structures, which appear to be consistent with previous findings describing laminal membrane structures (LMS) in murine primary neurons (Figure 5e) [21,22]. This phenomenon can be found in other regions of human neurons as well, recapitulating the formation of LMS in human neurons by POWV (Figure 5f). While the uninfected MAP2-stained co-cultures showed puncta which resembled the ones observed in POWV-infected cells, measurements of these puncta were found to be larger and have a wider size distribution in infected cells (Figure 5g,h and Appendix A). The mean size of the aberrations was found to be significantly larger at 5.3 µm, compared to the mean size of the puncta in MOCK neurons at 2.3 µm.

### 3.5. Powassan Virus Causes Aberrations in Mouse Brains

The formation of LMS in TBEV-infected murine primary neurons in vitro has previously been reported [21]; however, no published study yet has evaluated if formation of LMS also occurs in TBEV infection in vivo. We utilized tissue clearing and whole organ imaging to test if intracellular LMS can be detected in POWV-infected mouse brains. Mouse models for POWV infection have been established and result in lethal encephalitis starting between 6–8 days post infection [20]. BALB/c mice were infected with POWV via the intraperitoneal route and brains extracted at time of euthanasia. After inactivation with 10% formalin, brain hemispheres were cleared and stained, followed by two-photon microscopy. Three-dimensional renderings of the tissue showed widespread infection identified by positive anti-POWV signal (red signal in Figure 6a). This included areas previously been described to be infected by POWV [20,37,38], such as the olfactory bulb (Figure 6b), cortex (Figure 6c) and hippocampus (Figure 6d). This is also significant as it is, to the best of our knowledge, the first time that tissue clearing and three-dimensional reconstructions have been generated using flavivirus-infected brains. Deep tissue imaging of cleared brains was performed using two-photon microscopy and revealed punctate structures within the dendrites of POWV-infected brains (Figure 6e–h), which were not seen in uninfected control tissues (Figure 6i,j). These findings suggest that the aberrations observed in the hNSC-derived neuron/astrocyte co-culture model can also be observed in vivo and are congruent with the appearance of LMS from previous studies, although further experiments would be needed to confirm the presence of LMS.

## 4. Discussion

Tick-borne flaviviruses are widespread and can result in severe pathology, yet the underlying molecular causes of neuropathogenesis remain understudied. Previous studies have established small rodent animal models to study POWV-induced pathogenesis, but there is still a gap in knowledge in the understanding of POWV pathogenesis in human cells [20]. While several studies have utilized primary human neuronal cell systems, no defined cellular responses to POWV infection have been reported [14,23,24,29,33,39,40]. Recent studies have identified LMS to be formed in TBEV-infected murine primary cell culture, but the presence in human neurons or more importantly in vivo, have not yet been performed [21,22]. This study aimed to characterize cellular responses of a hNSC-derived neuron/astrocyte co-culture system to infection with POWV and also to lay the groundwork for defining the presence of LMS in human cells and in vivo.

### 4.1. Neuron/Astrocyte Co-Cultures Are Susceptible to WNV and POWV Infection

Neurotropic flaviviruses show a preference for neurons; however, they can also infect other neuronal cell types [17]. Our study demonstrates that hNSC-derived neuron/astrocyte co-cultures are susceptible to infection with both WNV and POWV and support viral replication. This is congruent with previous studies that show that both cell types are susceptible to infection by MBFVs and TBFVs [15,16,17]. We further describe that in addition to previously used differentiated stem cell systems, TBFVs can establish a long-term infection and are not directly responsible for cell death [14].

### 4.2. POWV Infection Elicits a Weak Cytokine Response

WNV- or POWV-infected neurons and astrocytes in vivo result in stimulation of pathogen recognition receptors (PRRs), such as protein kinase R (PKR) and toll-like receptors (TLRs) [16,41,42]. The stimulation of PRRs leads to a pro-inflammatory state where infected cells undergo apoptosis and further perpetuate the inflammatory response as well as recruit lymphocytes [16]. Our findings suggest that, in contrast to WNV, a reduced inflammatory response occurs in POWV-infected neuronal cells, which ultimately results in less apoptosis. However, increased levels of released chemokines could contribute to the recruitment of immune cells, particularly CD8+ lymphocytes, which drive the pathology of TBFVs described in animal models and clinical cases [11,12,24,36,43,44]. The combined data from our studies with previously published cytokine profiles and in vivo data suggests that pathogenesis from POWV encephalitis is not primarily an inflammatory response, but rather suggests that the recruitment of activated lymphocytes results in neuronal destruction, ultimately contributing to neuropathology.

### 4.3. Inflammatory Response to WNV Progresses Neuropathology

In contrast to POWV infection, a robust inflammatory response in WNV-infected hNSC-derived neuron/astrocyte co-cultures could be determined, including caspase cleavage and apoptosis. The apparent lack of apoptosis in POWV infected cells is expected, since no released inflammatory cytokines could be detected. While TBEV infection of neuron/astrocyte co-cultures produced inflammatory cytokines, it is noted that the neuronal survival was not affected by TBEV infection [14]. The clinical signs of POWV infection in animal models suggest a robust inflammatory response and CNS damage [20]. However, the absence of inflammation in our human hNSC-derived neuron/astrocyte co-cultures is at odds with this. We hypothesize that the lack of different glial cells, such as microglia or oligodendrocytes, combined with the absence of a peripheral inflammatory response might be responsible for the dampened cytokine response. High levels of IL-1β measured in WNV infected primary cells coupled with the secretion of high levels of pro-inflammatory cytokines (such as TNF-α, IL-6 and IL-8) are consistent with damaging inflammation that results in the destruction of neurons and astrocytes through apoptosis [16]. This is in line with current findings as WNV is known to cause a severe neuroinflammation in cell cultures and animal models [16,35,42]. Human cases are also characterized by production of pro-inflammatory cytokines in response to WNV infection [45,46]. This includes formation of inflammasomes, activation of the NF-κB pathway and ultimately apoptosis [47,48,49]. In addition, we noted an elevated level of IFN-γ in WNV infected co-cultures compared to POWV infected cells. This is important, as IFN-γ is often viewed as a marker for inflammation, and higher levels are associated with a robust innate immune response. The inflammation resulting from WNV has been described in animal models [35,45,50], and is a potent driving factor for both WNV encephalitis prior to recruitment and activation of lymphocytes. Previously, we could demonstrate that a robust innate immune response of hNSC-derived neuron/astrocyte co-cultures occurred to infection with La Crosse bunyavirus, as well as ZIKV [17,23]. Our data therefore suggest that the lack of POWV-induced inflammatory response or apoptosis might be a characteristic intrinsic feature to TBFVs.

### 4.4. Infection of Astrocytes by POWV Can Serve as a Cellular Viral Reservoir

The infection of astrocytes by TBFVs has been debated, as there is evidence that astrocytes are readily infected but also play a reduced role for infection in vivo [17,26,33,51]. Our studies suggest that while astrocytes can be infected by POWV, they are not a major source of inflammation. Astrocytes can serve another function in perpetuating viral infection though, by tolerating a low level of replicating virus and slowly releasing virus to infect surrounding cells and therefore become a reservoir for prolonged infection [26,51]. Previous studies have shown that rat astrocytes can be successfully infected and have a great resiliency to changes induced from TBEV [33]. Primary rat astrocytes have also been observed to be infected for up to 14 days without inducing cell death [26]. This tendency for rat astrocytes to be perpetually infected and serve as a viral reservoir is recapitulated with our studies in that human astrocytes in hNSC-derived neuron/astrocyte co-cultures are not a source of inflammatory cytokines or apoptosis and remain infected after 72 h of infection. One of the hallmarks of TBFV encephalitis is a tendency to cause long-term deficits and recurrent symptoms [38,52,53]. The long-term infection of non-neuronal cells could be a contributing factor to the long-term sequelae observed in POWV encephalitis in conjunction with the damage caused by lymphocytic infiltration. Currently, no approved therapeutic strategies exist for the treatment of TBFV infections. The established hNSC-derived neuron/astrocyte co-culture could potentially be utilized in future approaches to characterize the antiviral activity of small molecules targeting infected neurons and/or astrocytes.

### 4.5. Tick-Borne Flaviviruses form Laminal Membrane Structures

The confocal imaging of POWV-infected hNSC-derived neuron/astrocyte co-cultures yielded an unexpected discovery, namely the formation of aberrations, characterized as focal viral accumulations in the dendrites of the infected neurons. The distended areas in the neurons were found in the neuronal projections rather than the body or hillock of the neuron and closely resembled previously reported findings in murine primary neurons [21]. In murine primary neuron cultures, the formation of multi-walled LMS has been described and demonstrated to be the result of the stem-loop 2 region of the 5′ untranslated region (UTR) of TBFV genome binding to host proteins [21,22]. The protein implicated in the formation of the LMS is the FMR1 protein. However, this has only been described in murine primary cells with TBFVs other than POWV [21,22]. There is a high degree of conservation between human and murine FMR1 and there is absolute conservation among the stem-loop2 region of TBFV 5′ UTRs, therefore, it would be possible that the formation of LMS is occurring in human primary neurons as a result of POWV infection. Future studies will be required to confirm the formation of these structures, with electron microscopy being the standard [54].

The formation of punctate structures within the tissues of mice infected with POWV also supports the formation of LMS in vivo. If the structures observed in this study are LMS, they could be playing a contributing role to the neuropathological changes caused by TBFVs. The aberrations described in vivo and in cultures were not located in cells that were undergoing apoptosis, as suggested by the microscopy data and supported by the BioPlex data. This suggests a scenario where POWV infection is somewhat tolerated by neurons and can result in infected neurons not triggering apoptosis while also containing replicating virus. Previous studies have shown that there is a diminished ability for neurons to properly transport mRNA when LMS are present [22]. If the structures identified in our study are LMS, the infected animals may have a deficiency in neuronal signaling which is present in human cells as well.

### 4.6. Tissue Clearing Provides Advantages over Immunohistochemistry

The visualization of infected tissues has traditionally been performed using immunohistochemistry (IHC) or immunofluorescence assays (IFA) [11,20]. These methods are beneficial in that they can display virus distribution, infected cell types and pathology resulting from the infection. However, the drawbacks to these methods stem from the sectioning of the tissue and the inability to visualize larger tissue samples. We have demonstrated that applying tissue clearing followed by two-photon microscopy is an effective way to visualize the spatial distribution of POWV and the intercellular aspects of deep tissue imaging without having to section the tissue. We will be able to use the long-term fixation and optical clearing protocol that was developed in this study for adapting to future developments in tissue clearing. This is ideal for investigating infection in multiple tissue types, not just the brain.

In summary, the findings from this study indicate that POWV is able to evade the inflammatory response and apoptosis in hNSC-derived neuron/astrocyte co-cultures as well as forming structural aberrations in the dendrites of neurons in vitro and in vivo. The findings with POWV are contrasted to WNV, which causes inflammation and cell destruction in neuron/astrocyte co-cultures and shows no signs of structural anomalies.

## Figures and Tables

**Figure 1 pathogens-11-01218-f001:**
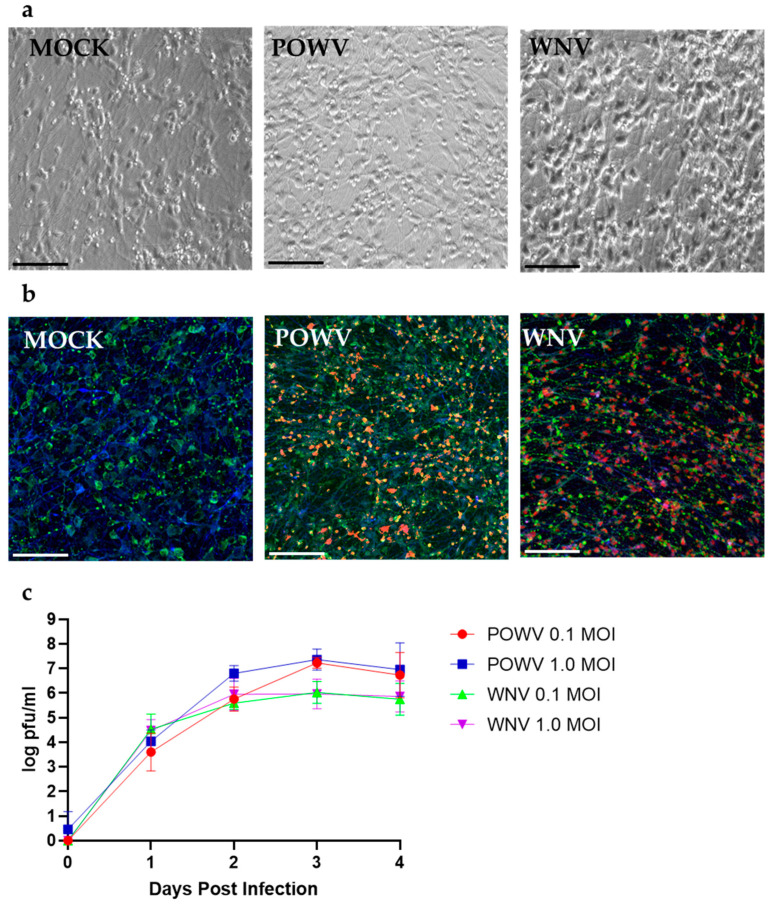
Susceptibility and replication kinetics for POWV and WNV in hNSC-derived neuron/astrocyte co-cultures. (**a**) Co-cultures were MOCK-infected or infected with 1.0 MOI of POWV or WNV and imaged at 72 h post infection via bright field microscopy. Cytopathic effect was determined by visually comparing mock and infected cells. (**b**) Identification of POWV- and WNV-infected cells in co-cultures. Cells were fixed after 72 h and then stained for neurons (MAP2, green), astrocytes (GFAP, blue), or virus (WNV or POWV, red) and imaged using confocal microscopy at 20×. (**c**) Growth kinetics for POWV and WNV were determined by collecting supernatant at different timepoints post-infection and tittered via plaque assay. Average data from biological triplicates are shown. Scale bars represent 50 µm.

**Figure 2 pathogens-11-01218-f002:**
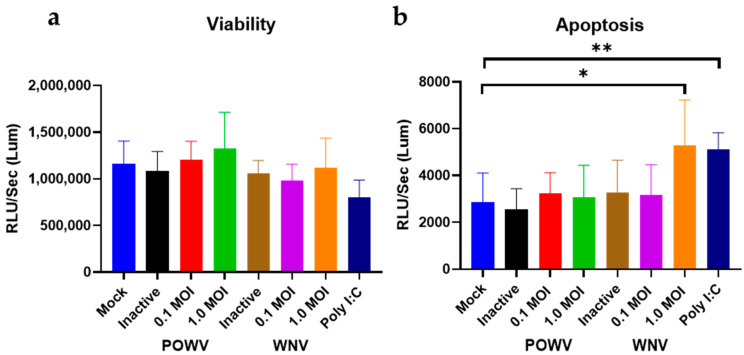
Cell viability and apoptosis in POWV- and WNV-infected hNSC-derived neuron/astrocyte co-cultures. Co-cultures were infected with either 0.1 MOI or 1.0 MOI of POWV or WNV and assayed for (**a**) cell viability or (**b**) apoptosis. After 72 h, assays were performed using a ProMega ApoTox/Glo kit and measured on a BioRad Cytation 7 reader. Poly I:C treatment was included as positive control, heat-inactivated virus as negative control. Infected samples and poly I:C samples were compared to the mock samples using an unpaired t-test with Welch’s correction. Average data from biological triplicates are shown. * *p* ≤ 0.05, ** *p* ≤ 0.01.

**Figure 3 pathogens-11-01218-f003:**
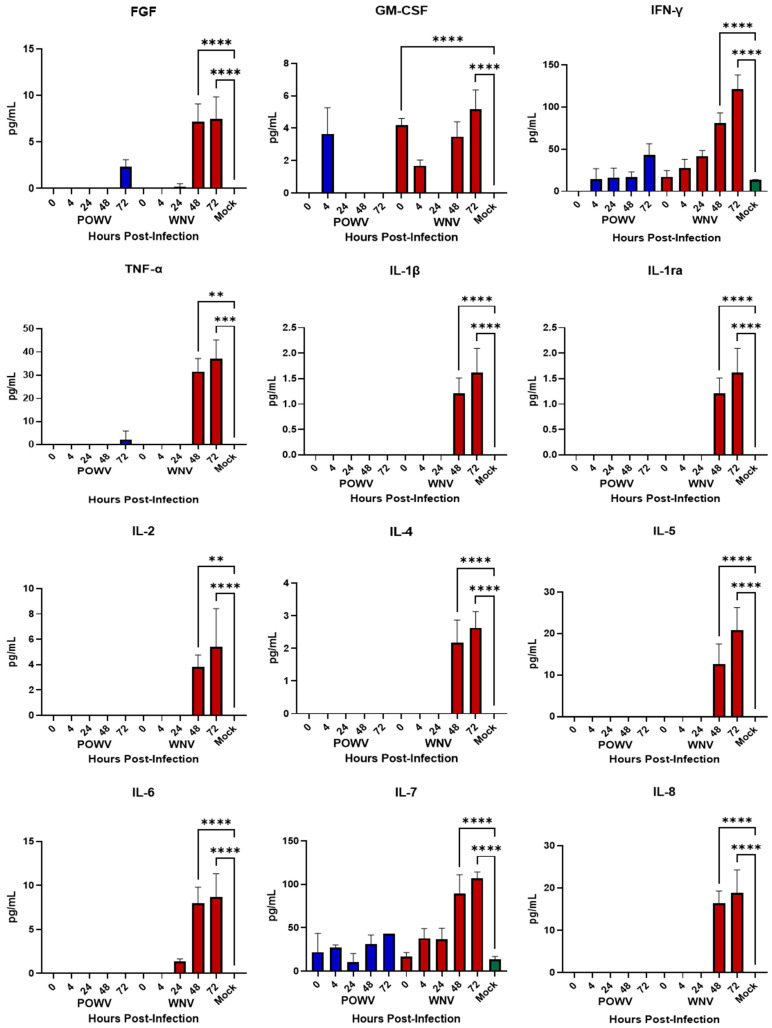
Cytokine and Chemokine Responses of hNSC-derived neuron/astrocyte co-cultures after infection with POWV or WNV. Differentiated hNSC co-cultures were infected with 1.0 MOI of POWV (blue bars) or WNV (red bars), supernatant aliquots sampled at 0, 4-, 24-, 48- or 72-h post-infection and analyzed via BioPlex for pro-inflammatory cytokines and chemokines. Green bars represent Mock data. Average data from biological data are shown. ** *p* ≤ 0.01, *** *p* ≤ 0.001, **** *p* ≤ 0.0001.

**Figure 4 pathogens-11-01218-f004:**
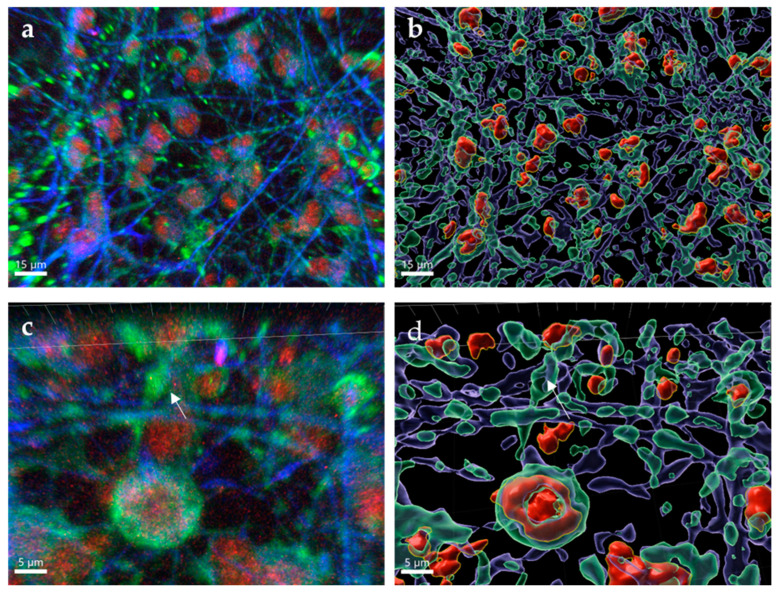
Characteristics of WNV infection in hNSC-derived neuron and astrocytes. hNSC-derived neuron/astrocyte co-cultures were infected with 1.0 MOI WNV and cells fixed at 72 h post-infection. Neurons were stained using anti-MAP2 antibodies (green), astrocytes were stained using anti-GFAP antibodies (blue) and WNV was stained using anti-E protein antibodies (red). (**a**) 63× optical image. (**b**) Three-dimensional rendering using IMARIS. (**c**) 63× optical image with 3× digital zoom. (**d**) Three-dimensional rendering using IMARIS. Note the presence of WNV (red) within the soma of neurons and astrocytes. In (**b**,**d**), cells were surface-rendered into translucent cells, with neurons represented by green surfaces, astrocytes by purple surfaces and WNV E protein by red surfaces. Arrows indicate fractured dendrites.

**Figure 5 pathogens-11-01218-f005:**
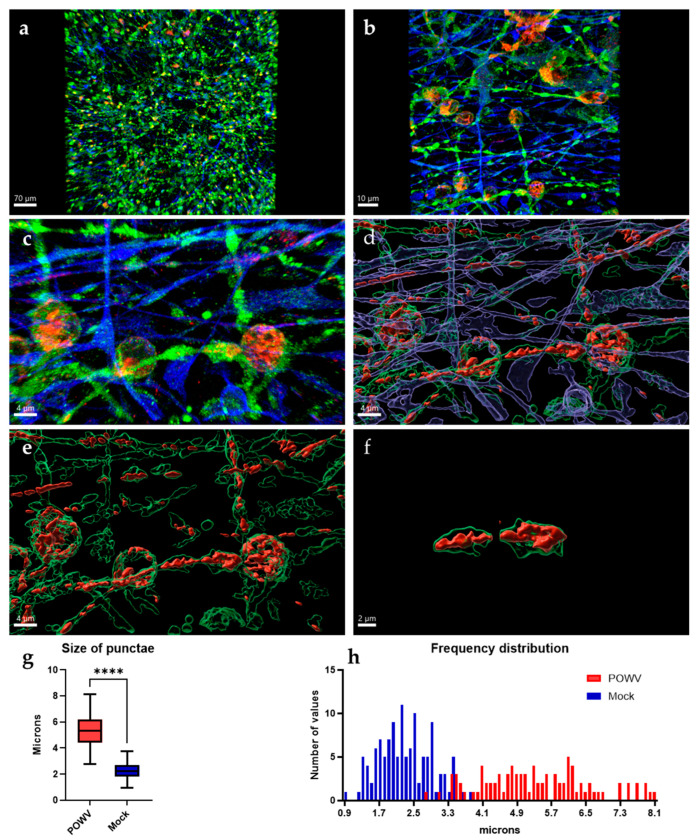
Characteristics of POWV infection in hNSC-derived neuron/astrocyte co-cultures. hNSC-derived neuron/astrocyte co-cultures were infected with 1.0 MOI POWV and cells fixed at 72 h post-infection. Neurons were stained using anti-MAP2 (green) antibodies, astrocytes were stained using anti-GFAP (blue) antibodies and POWV was stained using anti-E protein antibodies (red). (**a**) 20× optical image. (**b**) 63× optical image. (**c**) 63× optical image with 5× digital zoom. (**d**–**f**) Three-dimensional rendering of neurons and virus using IMARIS, focusing on viral aberrations. Average size of the aberrations and artifacts were compared by measuring the size of the MAP2 signal and (**g**) using a *t*-test with Welch’s correction and (**h**) size distribution was calculated in a histogram. **** *p* ≤ 0.0001.

**Figure 6 pathogens-11-01218-f006:**
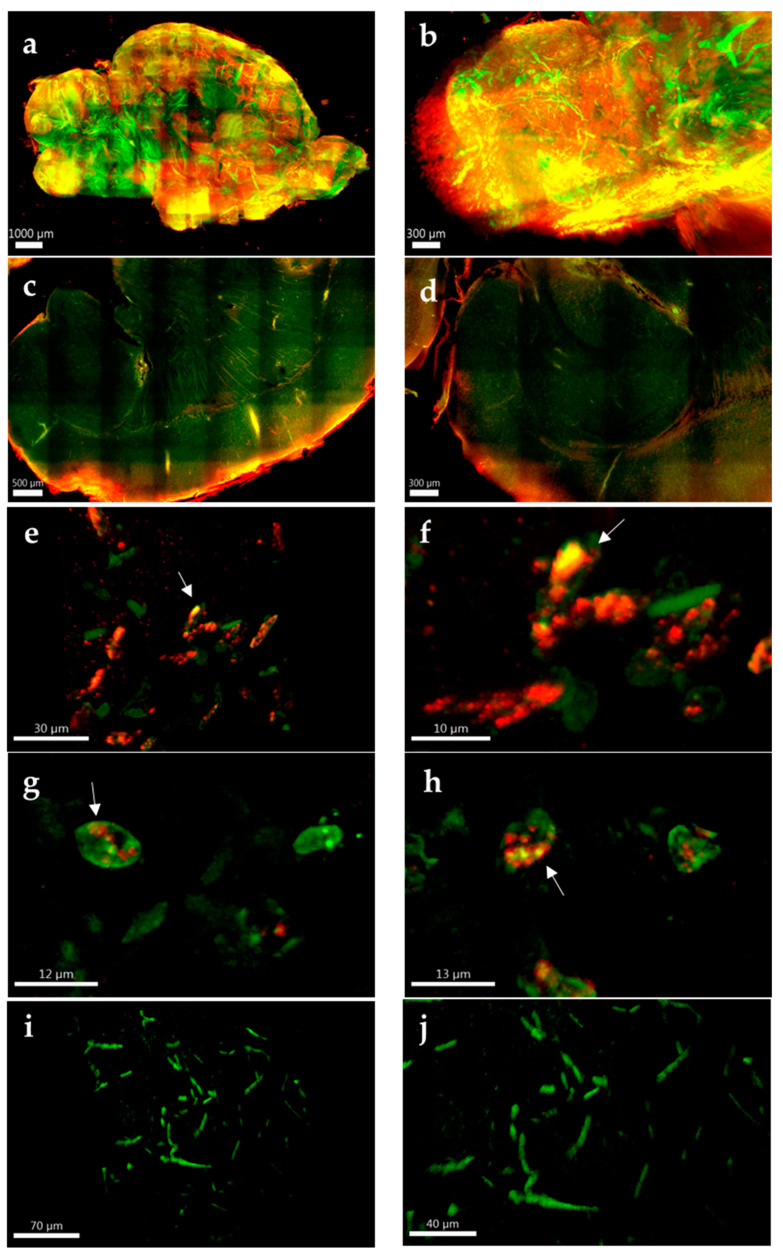
Whole and deep tissue imaging of POWV-infected mouse brains. BALB/c mice were infected with POWV, and brains collected at time of euthanasia. Brain hemispheres were fixed in 10% formalin, cleared and labeled with rabbit polyclonal anti-POWV (red) and chicken anti-MAP2 (green). (**a**) Two-photon microscopy with a 10× optical objective was used for three-dimensional reconstructions of the entire brain hemispheres. (**b**) Olfactory bulb, (**c**) cortex and (**d**) hippocampus are zoomed for detail. Two-photon microscopy using a 40× optical objective with digital zoom up to 5× was used for deep tissue imaging of (**e**–**h**) POWV-infected and (**i**,**j**) uninfected hemispheres. Arrows in (**e**–**h**) are highlighting possible LMS.

## Data Availability

No further datasets are available.

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
