# Peer review of "Powassan Virus Induces Structural Changes in Human Neuronal Cells In Vitro and Murine Neurons In Vivo"

_pathogens, 2022, doi:10.3390/pathogens11101218_

Round 1

Reviewer 1 Report

Nelson et al., determined apoptosis and cell viability in POWV- and WNV-infected hNSC-derived human neuron/astrocyte co-cultures which differentiated from human fetal neural stem cells. They measured the produced interleukins, and studied the intracellular histological changes, by indirect immunfluorescence with mono- and policlonal antibodies.

They confirmed the presence of LMS (caused also by other TBFV) in vitro in human neurons and astrocytes, and punctate structures (a possible pre-LMS) in dendtrites of in vivo infected mouse brain neurons. The intracellular lesions were restricted to (and also POWV) in dendrites, while WNV was present in the cell body. WNV produced a productive infection, caused much profound intracellular destruction both in astrocytes and neurons. Several differences between intracellular changes and immune response against of POWV and WNV were shown.

The work used appropriate well-documented methods for the histological studies, containes novel data about POWV histology and comparison of POWV and WNV a TBFV and a MBFV.  

sometimes comma before „and”, sometimes not

184 line- intraperitoneally not interperitoneally

-          Why did the authors sacrifice all teh mice on p.i.d. 8? 2-10 days would not have been betetr to examine the development of the elsions?

-          Animal experiment permission was not necessary?

Reviewer 2 Report

The topic will be interesting study for readers. The methods used in the study and preferred cell lines can be experienced for similar viruses.

Reviewer 3 Report

1-      The research was well planned and presented. If it is possible the authors could shorten the keyword “Human neural stem cell-derived neuron astrocyte co-culture”

2-      Line 41-42: The authors used too old a reference for Reference 1 from 1959 and there was no information about Flaviviridae.  It may be replaced by a recent publication.

3-      Line 51-53: Please could you add a reference for POWV case numbers? If you share it readers could reach easily data or change it according to official numbers. I found the case number belongs to 2011-2020 (n:178) on the CDC website (https://www.cdc.gov/powassan/statistics.html ).

4-      Line 54-56: “(case fatality rate 15%)”.  According to the CDC website POWV data (2011-2020); case fatality is 11% (https://www.cdc.gov/powassan/statistics.html ). Some of your references write approximately 10%. It could be changed according to the last mortality numbers.

5-      Line 61-63: Please check the references 10-11 according to your sentences. Check other references in the manuscript so readers could easily reach the relevant ones.

6-      Line 113, BHK-SA: Please write abbreviations long first in the manuscript.

7-      Could you provide the catalog numbers of all you used and mentioned in the material-methods section? So, other researchers could use your method.

8-      Line 130-132: Could you write which cell cultures were used for which virus (WNV and POWV)  inoculations?

9-      Line 188: “according to established protocols” Could you add references for these protocols? Researchers could reach them easily.

1-   There are a lot of “Data not shown” in the manuscript.  For example line 296, 299. Could you add supplementary data or revise them according to Pathogens instruction for authors:

Unpublished Data

Restrictions on data availability should be noted during submission and in the manuscript. "Data not shown" should be avoided: authors are encouraged to publish all observations related to the submitted manuscript as Supplementary Material. "Unpublished data" intended for publication in a manuscript that is either planned, "in preparation" or "submitted" but not yet accepted, should be cited in the text and a reference should be added in the References section. "Personal Communication" should also be cited in the text and reference added in the References section. (see also the MDPI reference list and citations style guide).

-The authors used   *** * symbol in the Figure 2 and Figure 5 but they didn't explain what the symbols meant in the figures.

Reviewer 4 Report

It is a very well written article.  It is only necessary to provide information about its use in vaccine and drug studies.

Reviewer 5 Report

In this study, Nelson and colleagues have developed a primary human fetal brain neural stem cell system to investigate the effect of infections by Powassan virus (POWV) and West Nile virus (WNV). They report that WNV infection stimulates production of inflammatory cytokines and activation of caspases 3 and 7, but POWV does not. They further show that POWV produces structures in dendrites in cultured cells and in brains of infected mice. 

General comments

The study is generally well done with a few experimental weaknesses as noted below. Most of the comments are related statements that need to be reworded for clarity. The major issues are:

1.     There are several statements that go beyond what the authors show in their experiments. These statements should be scaled back. 

2.     If possible, the authors should include images of mock-infected cells processed the same way as infected cells in Fig. 5. A better explanation is needed for why the authors believe the structures in uninfected cells are artifacts, whereas those in infected cells are not. 

3.     The authors state there is widespread cell death in WNV-infected cells, but their data in Fig 2A shows no significant effect on cell viability

4.     There seems to be other potential explanations for the activation of caspe3/7 in WNV-infected cells beyond the production of inflammatory cytokines.

Specific comments

Genus and species should be italicized.

Lines 52-53: “…this number is likely lower than the actual case number of infections due to the need for increased surveillance.” This statement is confusing. It seems like they authors are saying that a lack of surveillance is likely to lead to an underestimate of the number of cases, but this is an awkward way to say that.

Catalog numbers for key reagents should be provided, especially antibodies. In some cases, there are multiple related items available from the suppliers, which makes it difficult for the reader to reproduce the conditions used in this manuscript. Also, new antibodies could be developed by the suppliers in the future, so a reader would not be able to know if they are using the same reagent

There are multiple instances in which the authors state that a technique was performed as previously described but do not provide a citation. Please add citations for such cases.

Lines 176-180: common names for animals do not need to be capitalized

Lines 200-201: Please reword this sentence. It appears to be a run on sentence. 

Line 232: Delete or complete the sentence that starts with “Imaging depth of”

Line 257: “Analysis” should be “analyses”

Line 263: Should be “in culture. However, …”

Line 274: Reword this sentence. It is not clear if “infected cells were visually indistinguishable from each other” refers to the neurons and the astrocytes or to cells infected with POWV and WNV

Line 312: Titrated should be titered.

Line 318-319: Should be “There were no significant changes in the viability. However, …”

Line 326-327: The authors state that "… POWV is not causing direct cell destruction or apoptosis of infected neurons or astrocytes, while WNV is causing cell death.” Although the data demonstrates that caspase 3/7 have been activated in WNV-infected cells at an MOI of 1 (Fig 2b), there seems to be no significant effect on cell viability in Fig 2A. Please explain how this data shows that WNV is causing death.

Line 340: should be “both POWV and WNV, …”

Lines 361-363: The authors state that “These findings correlate well with the apoptosis data for WNV, signifying that an inflammatory response might be driving caspase cleavage, while a reduced inflammatory response might be tied to lower apoptosis, as observed for POWV.” Can the authors distinguish this possibility from other explanations? For example, WNV replication might directly activate the apoptotic pathway, but POWV replication does not? Alternatively, replication of both viruses might activate the apoptotic pathway, but POWV is able to inhibit the pathway prior to caspase3/7 activation. Do the authors know if the concentrations of cytokines produced in cultures of WNV-infected cells sufficient to activate caspase 3/7 in uninfected cells?

Figure 4: Are three-dimensional images rendered with Imaris available from mock-infected cells? If so, it would be helpful to the reader if these were included. 

Lines 412-413: The authors state that “The uninfected samples showed artifact from MAP2 staining that resembled punctae.” Images of uninfected samples should be included in Fig. 5 showing these structures so that the reader can better evaluate the results. Also, how do the authors know that the punctae in the uninfected cells are artifacts, but the structures observed in infected cells are real? This seems to be a significant issue that must be addressed.

Line 470: Subheading states that “Chemokines Contribute to POWV Pathogenesis More Than Cytokines.” This statement goes beyond what their experiments show and should be revised. The experiments here have not looked at the pathology of POWV infection in detail and have not conclusively established a role for chemokines in POWV pathology. The statement needs to be toned down.

Line491-493, legend for Fig.6: Why are panels i and j described before e-h?

Line 507: “indicative of” is too strong. Better to say “consistent with”

Line 550-551: This statement is vague. The authors should be more specific about what is meant by “homology between human and murine FMR1 and there is conservation among TBFV 5’ UTRs.” How much homology? Are human and murine FMR1 known to have the same binding specificity? Are the 5’ UTRs of TBFVs absolutely conserved? What about SL-2?

Lines 584-585: “causes widespread inflammation and cell destruction in human/astrocyte co-cultures.” This statement seems too strong. Fig 4, which showed no difference in cell viability, does not support the conclusion that there was widespread cell destruction.

Line 581, “summation” should not be capitalized
